# AutoNF: Automated Architecture Optimization of Normalizing Flows Using a Mixture Distribution Formulation

## Abstract

Although various flow models based on different transformations have been proposed, there still lacks a quantitative analysis of performance-cost trade-offs between different flows as well as a systematic way of constructing the best flow architecture. To tackle this challenge, we present an automated normalizing flow (NF) architecture search method. Our method aims to find the optimal sequence of transformation layers from a given set of unique transformations with three folds. First, a mixed distribution is formulated to enable efficient architecture optimization originally on the discrete space without violating the invertibility of the resulting NF architecture. Second, the mixture NF is optimized with an approximate upper bound which has a more preferable global minimum. Third, a block-wise alternating optimization algorithm is proposed to ensure efficient architecture optimization of deep flow models.

## 1 Introduction

Normalizing flow (NF) is a probabilistic modeling tool that has been widely used in density estimation, generative models, and random sampling. Various flow models have been proposed in recent years to improve their expressive power. Discrete flow models are either built based on elemental-wise monotonical functions, named autoregressive flow or coupling layers (Papamakarios et al., 2017), or built with transformations where the determinant of the flow can be easily calculated with matrix determinant lemma (Rezende & Mohamed, 2015). In the continuous flow family, the models are constructed by neural ODE (Grathwohl et al., 2019).

Despite the variety of flow models, there's yet no perfect flow concerning the expressive power and the computation cost. The flow models with higher expressive power usually have higher computational costs in either forward and inverse pass. In contrast, flows which are fast to compute are not able to model rich distributions and are limited to simple applications. For instance, autoregressive flows (Papamakarios et al., 2017) are universal probability approximators but are $D$ times slower to invert than forward calculation, where $D$ is the dimension of the modeled random variable $x$ (Papamakarios et al., 2021). Flows based on coupling layers (Dinh et al., 2015; 2017; Kingma & Dhariwal, 2018) have an analytic one-pass inverse but are less expressive than their autoregressive counterparts. Other highly expressive NF models (Rezende & Mohamed, 2015; Behrmann et al., 2019) cannot provide an analytic inverses and relies on numerical optimizations.

For different applications, the optimal flow model can be drastically different, especially if the computation cost is taken into consideration. For generative models (Dinh et al., 2015; Kingma & Dhariwal, 2018), flows with the fast forward pass are preferable since the forward transformations need to be applied to every sample from the base distribution. For density estimation (Papamakarios et al., 2017; Rippel & Adams, 2013), flows with cheap inverse will prevail. For applications where flow is utilized as a co-trained kernel (Mazoure et al., 2020), the computation cost and performance trade-off are more important, i.e., having a fast model with relatively good performance. However, in the current body of work, the architecture designs of the flow models are all based on manual configuration and tuning. To this date, there is a lack of a systematic way that could automatically construct an optimal flow architecture with a preferred cost.

In this paper, we propose AutoNF, an automated method for normalizing flow architecture optimization. AutoNF has a better performance-cost trade-off than hand-tuned SOTA flow models based on a given set of transformations. Our approach employs a mixture distribution formulation that can search a large design space of different transformations while still satisfying the invertibility requirement of normalizing flow. The proposed mixture NF is optimized via approximate upper bound which provides a better optimization landscape for finding the desired flow architecture. Besides, to deal with exponentially growing optimization complexity, we introduce a block-wise optimization method to enable efficient optimization of deep flow models.

## 2 Related Work

**Normalizing Flows**: Various normalizing flow models have been proposed since the first concept in (Tabak & Turner, 2013). Current flow models can be classified into two categories: finite flows based on layer structure, and continuous flow based on neural ODE (Grathwohl et al., 2019). The finite flow family includes flows based on elemental-wise transformation (Papamakarios et al., 2017; Kingma & Dhariwal, 2018) and flows whose transformations are restricted to be contractive (Behrmann et al., 2019). In elemental-wise transformation flows, autoregressive flow and coupling layers are two major flavors and extensive work has been proposed to improve the expressive power of both flow models. In Huang et al. (2018), the dimension-wise scalar transformation is implemented by a sigmoid neural network, which increases the expressive power at the cost of being not analytically invertible. In Durkan et al. (2019), piecewise splines are used as drop-in replacement of affine or additive transformations (Dinh et al., 2015; 2017) and is the current SOTA flow model. Consequently many recent research efforts have been devoted to closing the gap of expressive power, albeit at the cost of more complex and expensive transformations. Moreover, there has been no quantitative trade-off analysis between the performance and cost among different flows.

**Neural Architecture Search**: Many algorithms have been proposed or applied for neural architecture search. For instance, reinforcement learning (Zoph & Le, 2017), genetic algorithm (Real et al., 2017; Suganuma et al., 2018; Liu et al., 2018), Monte Carlo tree search (Negrinho & Gordon, 2017) or Bayesian optimization (Kandasamy et al., 2018). However, these methods all face the challenge of optimizing on a large discrete space and can take thousand of GPU days to find a good architecture. To address this issue, DARTS (Liu et al., 2019) proposes to relax the search space from discrete to continuous and allows efficient differentiable architecture search with gradient method which could reduce the search time to a single GPU day while still producing the SOTA architecture. However, all current NAS methods focus on optimizing traditional neural network structures (CNN, RNN) and there has yet been any implementation on normalizing flow.

**Necessity for the Trade-off Between Performance and Cost:** Despite various transformations proposed in the literature, there is no perfect transformation with strong expressive power and low computational cost. Autoregressive flows have better expressive power, but the inverse computation cost grows linearly with data dimension. Coupling layers' inverse calculation is as fast as the forward pass, but their expressive power is generally worse than autoregressive flow with the same element-wise transformation. Even in the same autoregressive flow or coupling layer family, flows with different element-wise transformations have different performance and computation costs. For instance, additive or affine coupling layers (Dinh et al., 2017; 2015) have very fast forward and inverse calculation with limited expressive power while the flow in (Durkan et al., 2019) are highly expressive but are more demanding on computation. In most applications, it is necessary to find the best performance while minimizing at least one specific component of the cost. Unfortunately, the current design of flow models is empirical and therefore cannot ensure the optimal trade-offs.

## 3 Method

In this work, we aim to tackle the challenge of finding an optimal flow model for a given task via an automated architecture search algorithm.

**Assumptions:** In the remaining part of this paper, without losing generality, we assume that the transformation is properly modeled such that during the training process, only forward computation is needed. Under this assumption, when the flow model is used for density modeling (Durkan

et al., 2019), the forward calculation is the dominant computation. When the flow model is used for random sampling (Kingma & Dhariwal, 2018), the inverse calculation is computationally intensive. When the flow model is utilized as a module and trained together with other components, e.g., policy network in maximum entropy learning (Mazoure et al., 2020), the training cost of the flow model is an important consideration.

**Problem Definition:** Given a transformation set with $m$ options $\{T^1, T^2, ...T^m\}$, the goal is to construct an optimal flow model with $n$ layers of transformations from the set. The flow model $p_{NF}(\boldsymbol{x}; \boldsymbol{\theta}) = p_{T_1 T_2 ... T_n}(\boldsymbol{x}; \boldsymbol{\theta})$ should minimize the KL divergence between the target distribution $p^*(x)$ and itself while minimizing its computational cost $C_{NF}$. Here, $\boldsymbol{\theta}$ are the parameters of the transformation in the flow model. In this paper, we use the forward KL divergence as our target loss function (Papamakarios et al., 2021):

$$\boldsymbol{\theta}^* = \arg\min_{\boldsymbol{\theta}} \{D_{KL}[p^*(\boldsymbol{x}) \,||\, p_{T_1 T_2 ... T_n}(\boldsymbol{x}; \boldsymbol{\theta})] + \lambda \cdot C_{NF}\}$$
$$\text{s.t.} \quad T_i \in \{T^1, T^2, ...T^m\} \tag{1}$$

While $\lambda$ is a tuning factor capturing the relative importance of the performance-cost trade-off. Finding this optimal flow model is a discrete optimization problem with exponential complexity. To enable efficient architecture optimization, we use proposed method of relaxing the discrete search space to continuous space as suggested in Liu et al. (2019).

## 3.1 MIXED FLOW ENSEMBLE

For the $i_{th}$ transformation layer with $m$ options, we introduce a corresponding weight $w_i^j$ for each option $T^j$ which reflects how likely the transformation will be selected. The weight is parameterized by a vector $\boldsymbol{\alpha}$ and made continuous via softmax:

$$w_i^j = \frac{\exp(\alpha_i^j)}{\sum_{j=1}^m \exp(\alpha_i^j)} \tag{2}$$

By applying this parameterization for each transformation layer, we can construct a ***mixed flow ensemble*** $p_{Mix}(\boldsymbol{x}; \boldsymbol{\theta}, \boldsymbol{\alpha})$, where each layer in this mixed model reflects a weighted combination of the effect of all possible transformations. In this case, the architecture optimization problem is reduced to learning the weight vector for each layer and, at the end of the optimization process, weights will be binarized and the transformation with the highest weight in one layer will be selected as the final transformation. The mixed flow ensemble thus degrades to a normal flow model. The whole procedure is illustrated in Fig. 1 (left).

As adopted in (Liu et al., 2019), training of the flow ensemble becomes joint optimization of the architecture parameter $\boldsymbol{\alpha}$ and the model parameter $\boldsymbol{\theta}$ over the training and validation datasets, which could be written as the following bi-level optimization problem:

$$\boldsymbol{\alpha}^* = \arg\min_{\boldsymbol{\alpha}} \; D_{KL}^{val}[p^*(\boldsymbol{x}) \,||\, p_{Mix}(\boldsymbol{x}; \boldsymbol{\theta}^*, \boldsymbol{\alpha})] + \lambda \cdot C_{Mix}(\boldsymbol{\alpha})$$
$$\text{s.t.} \quad \boldsymbol{\theta}^* = \arg\min_{\boldsymbol{\theta}} \; D_{KL}^{train}[p^*(\boldsymbol{x}) \,||\, p_{Mix}(\boldsymbol{x}; \boldsymbol{\theta}, \boldsymbol{\alpha})], \tag{3}$$
$$\forall \, T \in p_{Mix}, \; T \in \{T^1, T^2, ...T^m\},$$

While the optimization problem is well defined, the key challenge is to construct the flow ensemble within the normalizing flow framework. This is different from traditional neural architecture search, which can mix various operations with no additional issue. Normalizing flow has its unique requirement for the invertibility of transformations and a preferred simple Jacobian calculation, which requires careful handling.

The mixed flow ensemble $p_{Mix}(\boldsymbol{x}; \boldsymbol{\theta}^*, \boldsymbol{\alpha})$ must satisfy two requirements. First, it must be a legal density function such that it can be optimized by the KL divergence formulation. Second, each transformation layer in $p_{Mix}(\boldsymbol{x}; \boldsymbol{\theta}^*, \boldsymbol{\alpha})$ should represent a weighted combination of all possible transformations. Consider the $i_{th}$ layer in the mixed flow ensemble with input random variable $\boldsymbol{x}_{in}$ and output random variable $\boldsymbol{x}_{out}$, and $p_{\boldsymbol{x}_{in}}(\boldsymbol{x}_{in})$ and $p_{\boldsymbol{x}_{out}}(\boldsymbol{x}_{out})$ are their corresponding density functions. This layer has $m$ transformation options in $\{T_i^1, T_i^2, ...T_i^m\}$ and $w_i^j$ is the corresponding

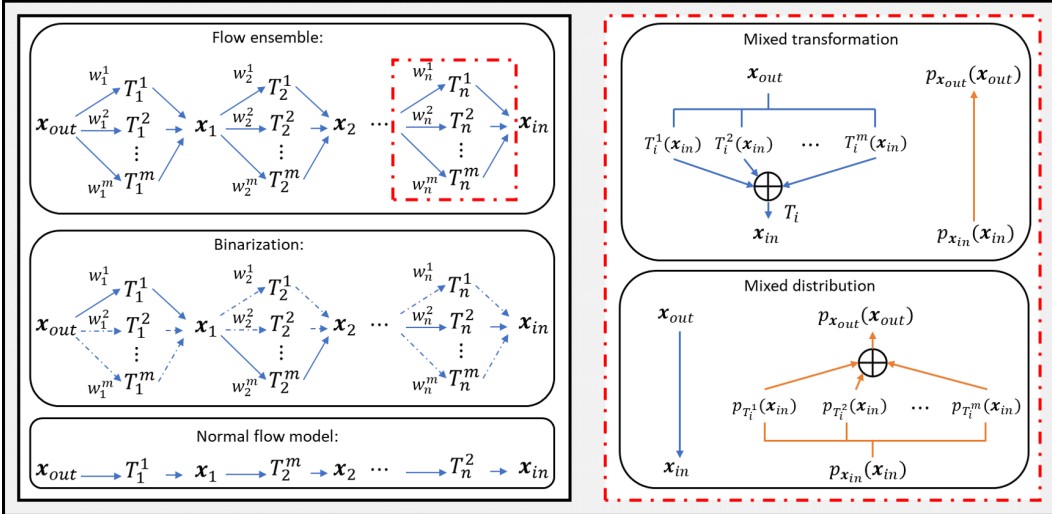

Figure 1: Left-top: the relaxation of search space and the flow ensemble is shown in Fig. 1. Left-middle: binarization of weights. Left-bottom: degradation to normal flow architecture. Right-top: construction flow ensemble by mixed transformations. Right-bottom: construction of flow ensemble by mixing distributions. The blue line in right indicates transformation on random variables and the orange line reflects change in distributions.

weight for each transformation. As discussed in Assumption, we assume all transformations directly model the inverse transformation, i.e. $\boldsymbol{x}_{in} = T_i^j(\boldsymbol{x}_{out})$. Two approaches can be used to construct the mixed flow ensemble.

**Construction by Mixed Transformations:** The straight forward way of building the $i_{th}$ mix flow ensemble layer is to mix all transformations by weighted summation, as shown in Fig. 1 (right-top). The final weighted transformation for this layer can be thus represented as:

$$T_i(\boldsymbol{x}_{in}) = \sum_{j=1}^{m} w_i^j \cdot T_i^j(\boldsymbol{x}_{out}) \tag{4}$$

There are two drawbacks of this formulation despite its simplicity. First, definition of normalizing flow requires the mixed transformation $T_i$ be invertible and differentiable in order to ensure $p_{\boldsymbol{x}_{out}}(\boldsymbol{x}_{out})$ legal density function. However, this invertibility is not guaranteed even if all candidate transformations are invertible. Second, even if the mixed transformation is invertible, there is no easy way to calculate the Jacobian determinant of this weighted summation of transformations. Meeting the requirement of invertibility and ease of calculating Jacobian determinant brings strict restrictions on the candidate transformations and prevents the optimization of flow architectures on a wider search space. As a result, the construction of the mixed flow ensemble by weighted summation of transformations is not adopted in this paper.

**Construction by Mixed Distributions:** An alternating way is to build the mixed flow ensemble by mixing distributions. For a given transformation $T_i^j$ in this $i_{th}$ layer, applying the transformation to the input random variable will result in a new distribution:

$$p_{T_i^j}(\boldsymbol{x}_{out}) = p_{\boldsymbol{x}_{in}}(T_i^j(\boldsymbol{x}_{out})) \cdot |\det \boldsymbol{J}_{T_i^j}(\boldsymbol{x}_{out})| \tag{5}$$

By applying this to every transformation option in $\{T_i^1, T_i^2, ...T_i^m\}$, we can obtain $k$ different distributions, and it is possible to mix all the density functions together by their weighted summation, to get a mixture model as shown in eq.(6).

$$p_{T_i(\boldsymbol{x}_{out})} = \sum_{j=1}^{m} w_i^j \cdot p_{T_i^j}(\boldsymbol{x}_{out}) \tag{6}$$

An illustration of this process is shown in Fig. 1 (right-bottom). Different from the previous approach, the mixture model has a legal density function as: $p_{T_i}(\boldsymbol{x}_{out})$. By the definition of normalizing flow, we can assume that there exists an invertible and differentiable transformation $T_i$, which transforms $\boldsymbol{x}_{in}$ to $\boldsymbol{x}_{out}$, although the transformation itself can not be explicitly written out.

For the next $(i + 1)_{th}$ layer, the density of the mixture model will be used as the input density function $p_{\boldsymbol{x}_{in}}(\boldsymbol{x}_{in})$ as in the previous layer. By applying this formulation for $n$ layers, the final mixed flow ensemble can be written as:

$$p_{Mix}(\boldsymbol{x}; \boldsymbol{\theta}, \boldsymbol{a}) = \sum_{k=1}^{m^n} W_k \cdot p_{T_1 T_2 ... T_n}(\boldsymbol{x}, \boldsymbol{\theta}) = \sum_{k=1}^{m^n} W_k \cdot p_i(\boldsymbol{x}; \boldsymbol{\theta}_i)$$

$$\text{where each} \quad W_k = \prod_{i=1}^{n} w_i \quad \text{and} \quad \sum_{k}^{m^n} W_k = 1 \tag{7}$$

Each $w_i$ is defined in eq.(2) and we use $p_k(\boldsymbol{x}; \boldsymbol{\theta}_k)$ to represent a "normal flow architecture" with $n$ transformation layers. Clearly, the final mixed flow ensemble is a legal density function which is in fact, a weighted summation of all possible flow models built with $n$ layers of transformations.

## 3.2 Optimization With Approximated Upper Bound

Optimizing the forward KL divergence between the target distribution and the mixed flow ensemble can be written as:

$$\mathcal{L}_{pMix}^{O} = D_{KL} \left[ p^*(\boldsymbol{x}) \,||\, p_{Mix}(\boldsymbol{x}; \boldsymbol{\theta}, \boldsymbol{\alpha}) \right]$$

$$= -\mathbb{E}_{p^*(\boldsymbol{x})} [\log(\sum_{k=1}^{m^n} W_k \cdot p_k(\boldsymbol{x}; \boldsymbol{\theta}_k))] \tag{8}$$

We will demonstrate that direct optimization of this original loss can lead to underside mixture models. In the whole search space of the flow ensemble, we are interested only in "normal flow architectures" points, i.e. the points where the weight of one architecture is 1 and others are all 0. However, it can be easily proven that the global optimum of $\mathcal{L}_{pMix}^{O}$ may not be the desired normal flow architecture (the red points in Fig. 2). Instead, optimization is very likely to end up in a mixture model that is globally optimal with similar weight for each possible flow architecture (the green point in Fig. 2). In this case, we will encounter difficulty when extracting a normal flow architecture with the search result. A common practice in differentiable architecture search (Liu et al., 2019) is to binarize the weights and select corresponding transformations. However, there is no guarantee that the binarized architecture will have a lower loss, and finding this nearest binarization may lead to performance drop. As a result, optimization with the original loss function is not suitable, and could be risky.

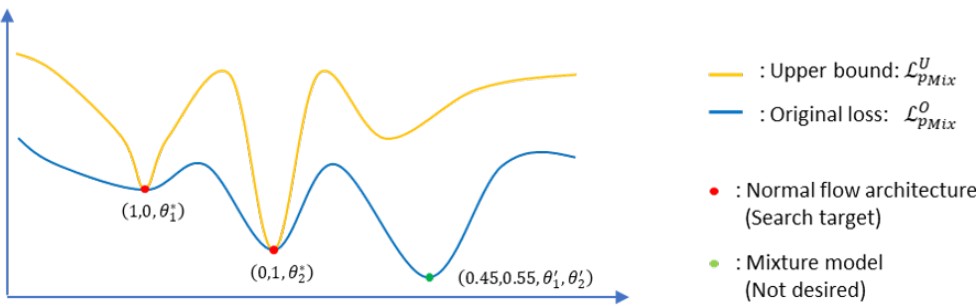

Figure 2: An illustrative example of the original loss and upper bound for a flow ensemble with 2 possible architectures. The red points indicate desired normal flow architectures and the green point indicates the global minimum of $\mathcal{L}_{pMix}^{O}$, which is a mixture model. The parameters $(a, b, \theta_1, \theta_2)$ refer to the weight of architecture 1, architecture 2 and their corresponding parameters.

In this paper, we propose to optimize an upper bound of the original loss function to provide a better global optimum for the search of best normal flow architectures. Our method utilizes Jensen's inequality $\log(\sum W \cdot x) \geq \sum W \cdot \log(x)$ as follows, since we have $\sum W = 1$ and the log function is concave, we can obtain an upper bound of the KL divergence given as:

$$\mathcal{L}_{p_{Mix}}^{O} = -\mathbb{E}_{p^*(\boldsymbol{x})}[\log(\sum_{k}^{m^n} W_k \cdot p_k(\boldsymbol{x};\boldsymbol{\theta}_k)] \leq \mathcal{L}_{p_{Mix}}^{U} = -\mathbb{E}_{p^*(\boldsymbol{x})}[\sum_{k}^{m^n} W_k \cdot \log(p_k(\boldsymbol{x};\boldsymbol{\theta}_k))] \quad (9)$$

The benefit of optimizing the upper bound can be summarized as follows:

**Proposition 1**: *The global minimum point of $\mathcal{L}_{p_{Mix}}^{U}$ is defined by a normal flow architecture.*

**Proof**: Suppose each flow model $p_k(\boldsymbol{x};\boldsymbol{\theta}_k)$ has an optimal parameter $\boldsymbol{\theta}_k^*$ that minimizes the KL divergence between $p^*(x)$ and it:

$$-\mathbb{E}_{p^*(\boldsymbol{x})}[\log(p_k(\boldsymbol{x};\boldsymbol{\theta}_k^*)] \leq -\mathbb{E}_{p^*(\boldsymbol{x})}[\log(p_k(\boldsymbol{x};\boldsymbol{\theta}_k)] \quad (10)$$

There also exists a flow architecture $(p_z(\boldsymbol{x};\boldsymbol{\theta}_z^*))$ that has the minimal KL divergence:

$$-\mathbb{E}_{p^*(\boldsymbol{x})}[\log(p_z(\boldsymbol{x};\boldsymbol{\theta}_z^*)] \leq -\mathbb{E}_{p^*(\boldsymbol{x})}[\log(p_k(\boldsymbol{x};\boldsymbol{\theta}_k^*)], \; \forall k \in m^n \quad (11)$$

We can then prove the proposition by showing that:

$$\mathcal{L}_{p_{Mix}}^{U} = -\mathbb{E}_{p^*(\boldsymbol{x})}[\sum_{k}^{m^n} W_k \cdot \log(p_k(\boldsymbol{x};\boldsymbol{\theta}_k))] \geq -\mathbb{E}_{p^*(\boldsymbol{x})}[\sum_{k}^{m^n} W_k \cdot \log(p_k(\boldsymbol{x};\boldsymbol{\theta}_k^*))]$$

$$\geq -\mathbb{E}_{p^*(\boldsymbol{x})}[\sum_{k}^{m^n} W_k \cdot \log(p_z(\boldsymbol{x};\boldsymbol{\theta}_z^*))] = -\mathbb{E}_{p^*(\boldsymbol{x})}[\log(p_z(\boldsymbol{x};\boldsymbol{\theta}_z^*)] \quad (12)$$

**Proposition 2:** *At normal architecture points ($W_k = 1$, $\boldsymbol{W}_{-k} = \boldsymbol{0}$), $\mathcal{L}_{p_{Mix}}^{U} = \mathcal{L}_{p_{Mix}}^{O}$.*

The proof of proposition 2 is apparent.

With the above propositions and under the assumption that the global optimum can be reached at the end of the optimization, we can show that the solution set, i.e. all possible normal flow architectures are the same in both $\mathcal{L}_{p_{Mix}}^{O}$ and $\mathcal{L}_{p_{Mix}}^{U}$, and we can do optimization with proposed upper bound without violating the original definition. Furthermore, since the global optimum of the upper bound will always lead to a normal flow architecture, we will not end up in finding a mixture model with the need to do heuristic and risky binarization of weights $W$.

## 3.3 Efficient Architecture Optimization for Deep Flow Models

While the flow ensemble by mixed density formulation could reflect the weighted effect of all possible transformation combinations, the architecture optimization complexity grows exponentially with respect to the number of considered transformation types and the number of transformation layers. In this scenario, efficient optimization of the whole flow architecture will not be possible. It is natural to decompose the original problem into sequential optimization of few different blocks, where each block could be optimized in one time with a limited number of layers. We propose two methods to decompose the problem.

**Grow Method:** The first approach is a straightforward greedy method which we call "Grow". Each time, a block is optimized until convergence, and the weights of the transformation layer are binarized. The searched transformations in this block will be directly added to the searched layer in the previous block. The architecture optimization of later blocks will be based on the existing layers and, the growth of layers stops when reaching the total number of layers constraint. Despite its simplicity, the downside of the "Grow" method is that the optimization is short-sighted. The block being optimized has no information about the architectures which could be added later, and the whole architecture is more likely to be trapped in local minimum.

**Block Method:** To avoid the issue of getting stuck in a local minimum, we propose another method named "Block" optimization. Blocks $\boldsymbol{B}$ in this approach are optimized alternatively to allow each block to adjust their architectures with respect to other blocks. In fact, the first "Grow" approach is a specific case of the "Block" method, where all the blocks are initialized as identity transformations and optimized only once.

---

**Algorithm 1** Algorithm flow for AutoNF

---

**Require:** Transformations: $\{T^1, T^2, ...T^m\}$, Blocks: $\boldsymbol{B} = \{B_1, B_2, ...B_l\}$, Cost: $C_{Mix}$
**Ensure:** $n$-layer flow model:
1: **while** not converged **do**
2:    **for** each $B_i \in \boldsymbol{B}$ **do**
3:       **while** not convergence **do**
4:          $\boldsymbol{\alpha}_{B_i} = \arg\min_{\boldsymbol{\alpha}_{B_i}} D_{KL}^{val}[p^*(\boldsymbol{x}) \,||\, p_{Mix}(\boldsymbol{x}; \boldsymbol{\theta}_{\boldsymbol{B}}^*, \boldsymbol{\alpha}_{B_i})] + \lambda \cdot C_{Mix}(\boldsymbol{\alpha}_{B_i})$
5:          $\boldsymbol{\theta}_{\boldsymbol{B}} = \arg\min_{\boldsymbol{\theta}_{\boldsymbol{B}}} D_{KL}^{train}[p^*(\boldsymbol{x}) \,||\, p_{Mix}(\boldsymbol{x}; \boldsymbol{\theta}_{\boldsymbol{B}}, \boldsymbol{\alpha}_{B_i})]$
6:       **end while**
7:       Fix architecture for $B_i$
8:    **end for**
9: **end while**

---

### 3.4 Cost Model and Algorithm Flow

As discussed in section II, we are interested in modeling the training cost (forward calculation cost) and the inverse calculation cost, since each of them plays a different role based on desired applications. We use an independent experiment to model the cost of different types of flows and summarized in a table which are included in Appendix B. With the cost model, the total cost of the mixed flow ensemble could be extracted based on emphasize on different costs, e.g. if training cost is the major concern, only training cost of different flows will be calculated. This total cost $C_{Mix}$ is then added as an regularization term into the training loss function.

In our paper, gradient based method is used for optimization which is efficient in this very high dimensional search space. The architecture parameter $\boldsymbol{\alpha}$ and the flow model parameter $\boldsymbol{\theta}$ are optimized alternatively with first order approximation in (Liu et al., 2019). The final algorithm flow of our proposed AutoNF method can be summarized in Algorithm 1.

## 4 Experiments

### 4.1 Evaluation of Proposed Upper Bound

**Setup:** We use a simple example to demonstrate the necessity of doing optimization with our proposed upper bound. We use AutoNF to build a 4 layer flow model with 2 transformation options including planar flow and radial flow from (Rezende & Mohamed, 2015). We use the POWER dataset as the target and optimize with original loss (name M1) and our proposed upper bound (named M2). We use Adam optimizer for both architecture parameter and model parameter with a learning rate of 0.002. The batch size is 512 and the training iteration is 10000.

The results are shown in Fig.3. For both M1 and M2, we present the weight for planar and radial flow for each layer as well as the training and validation loss during the search process. The final weight for each layer, searched architectures after binarization and the test score are shown in the right-bottom table.

**Analysis:** Optimization with our proposed upper bound (M2) shows a concrete convergence of weight to 0 or 1 for each layer, which leads to a desired normal flow architecture, while the optimization with the original loss function (M1) ends up in a mixture model instead of a normal flow architecture, as shown in Fig.3(left). This is within in our expectation as shown in Fig.2. Moreover, although the mixture model is mostly likely to be the optimal in the original loss, the normal flow architecture after binarization however, is not an optimal model. As shown in the right-bottom table, the architecture found by M2 has a significantly better test score than M1, and this clearly supports our statement of doing optimization with our proposed upper bound.

### 4.2 Search for flow Models with Best performance Cost Trade-off

**Transformation Options and Reference Designs:** To evaluate our AutoNF framework, we setup our experiments with four types of non-linear flows and one linear flow. In autoregressive family, we

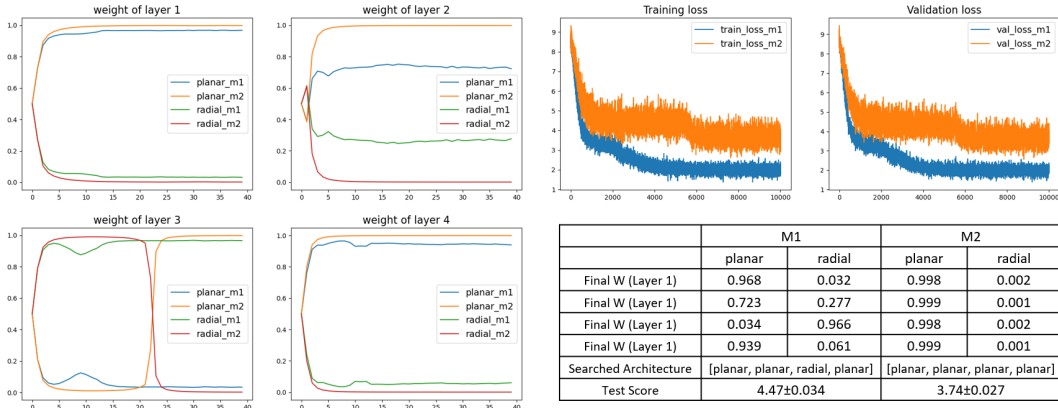

| | | M1 | | M2 | |
|---|---|---|---|---|---|
| | | planar | radial | planar | radial |
| Final W (Layer 1) | | 0.968 | 0.032 | 0.998 | 0.002 |
| Final W (Layer 1) | | 0.723 | 0.277 | 0.999 | 0.001 |
| Final W (Layer 1) | | 0.034 | 0.966 | 0.998 | 0.002 |
| Final W (Layer 1) | | 0.939 | 0.061 | 0.999 | 0.001 |
| Searched Architecture | | [planar, planar, radial, planar] | | [planar, planar, planar, planar] | |
| Test Score | | 4.47±0.034 | | 3.74±0.027 | |

Figure 3: The result of optimization of a 4-layer flow ensemble with transformation options between planar flow and radial flow with original loss and proposed upper bound. The left four figures are the weight for each layer during the search process. The right-top figures are the training and validation loss during training. The right-bottom table collects final weight for each layer, the searched architecture, and their test score (lower the better).

choose affine autoregressive flow (Papamakarios et al., 2017) and rational quadratic autoregressive flow (Durkan et al., 2019). Affine autoregressive flow has limited expressive power but the computation cost is lower, while the later has the state of art performance in autoregressive family with higher cost. Affine coupling layer (Dinh et al., 2015) and rational quadratic coupling layer (Durkan et al., 2019) are selected from coupling layer family. For linear transformation, we combine a reverse permutation and an LU linear layer together as a single layer. Random permutation (Durkan et al., 2019; Oliva et al., 2018) is not used since it is difficult to reproduce in architecture optimization. Every non-linear transformation layer is paired with a linear transformation layer suggested by Durkan et al. (2019) as a final transformation option, i.e., a layer in our experiment contains a reverse permutation, an LU-linear layer and one of the non-linear transformation layer listed above.

We use the rational quadratic flows family, including rational quadratic autoregressive flow (RQ-AF) and Rational quadratic coupling layer (RQ-C) in (Durkan et al., 2019) which have top 2 performance as the baseline. For fair comparison, we use RQ-AF as the baseline when emphasizing forward cost since it has better performance and use RQ-C as the baseline when emphasizing inverse cost since RQ-C has significantly lower inverse cost.

**Evaluation Metric and Datasets:**    Evaluating the performance-cost trade-off is an open question in NF, we propose to use a new metric to address the difficulty of negative log-likelyhood (NLL). NLL is a common measurement for density estimation (lower, the better), however, the order of magnitude of NLL is different across different datasets and it is not suitable to use percentage difference to measure how a model is exactly better than another.

In this paper, We proposed to utilize density and coverage (Naeem et al., 2020) to evaluate the performance of NF models. Density and coverage are recently proposed method to evaluate the sample quality of generative models. The density metric reflects the fidelity of the model and is consistent with NLL metric. Across different datasets, density and coverage are at the same order of magnitude and allows evaluation of architecture across datasets. In our experiments, 10000 samples are drawn from the trained flow models and compare with 10000 samples from the test data. The results of three independent runs are averaged as the final reported results.

To evaluate the performance-cost trade-off, we define a figure of merit (FOM) as FOM = cost reduction% + density drop% compared to reference SOTA designs. In principle, the weight of the two terms can be manually adjusted to reflect the importance. For demonstration purpose, we use the equally weighted summation to report the results.

The performance of the flow models are evaluated with density estimation for UCI (Dua & Graff, 2017) and BSDS300 (Martin et al., 2001) datasets.

**Analysis:** The architecture search results are reported in Table.1 which includes the test NLL, density, coverage, cost and corresponding FOM. Table.1 shows that our AutoNF clearly helps to find architectures that have better performance-cost trade-off. out AutoNF can reach to up to $3.66X$ cost reduction and up to $75.2\%$ improvement in FOM compared with SOTA literature results. Across all five different datasets, AutoNF demonstrates an average improvement of $58.67\%$ on FOM with emphasis on forward cost and an average improvement of $52.57\%$ on FOM with emphasis on inverse cost.

Table 1: Performance and cost trade-off between searched architectures and human designed architectures on UCI density estimation datasets. When the emphasizing on forward cost, the cost column reports forward cost, when emphasizing on inverse cost, the cost column reports inverse cost. The test NLL (lower the better), density, coverage (higher the better) cost and FOM are reported.

| Datasets | Cost Emphasize | Architectures | Test NLL | Density | Coverage | Cost | FOM |
|---|---|---|---|---|---|---|---|
| POWER | Forward | RQ-AF | -0.66±0.01 | 0.9909 | 0.9637 | 16.64 | 0 |
| | | AutoNF | -0.24±0.01 | 0.9677 | 0.9558 | 4.41 | +74.4% |
| | Inverse | RQ-C | -0.64± 0.01 | 0.9715 | 0.96 | 13.65 | 0 |
| | | AutoNF | -0.210.01 | 0.957 | 0.9547 | 3.73 | **+75.2%** |
| GAS | Forward | RQ-AF | -13.09 ± 0.02 | 0.7939 | 0.8982 | 16.64 | 0 |
| | | AutoNF | -9.12±0.02 | 0.6233 | 0.769 | 5.41 | +46.00% |
| | Inverse | RQ-C | -13.09 ± 0.02 | 0.7874 | 0.8965 | 13.65 | 0 |
| | | AutoNF | -8.45±0.03 | 0.5186 | 0.687 | 4.73 | +31.21% |
| HEPMASS | Forward | RQ-AF | 14.01 ± 0.03 | 0.9428 | 0.9588 | 16.64 | 0 |
| | | AutoNF | 17.36±0.02 | 0.9074 | 0.9549 | 6.49 | +57.23% |
| | Inverse | RQ-C | 14.75±0.03 | 0.7822 | 0.9556 | 13.65 | 0 |
| | | AutoNF | 19.32±0.02 | 0.8841 | 0.9471 | 5.73 | +54.84% |
| MINIBOONE | Forward | RQ-AF | 9.22 ± 0.48 | 0.9034 | 0.922 | 16.64 | 0 |
| | | AutoNF | 10.91±0.44 | 0.7957 | 0.9274 | 3.94 | +64.40% |
| | Inverse | RQ-C | 9.67 ± 0.47 | 0.8257 | 0.8982 | 13.65 | 0 |
| | | AutoNF | 13.56±0.57 | 0.6816 | 0.8768 | 3.73 | +55.22% |
| BSDS300 | Forward | RQ-AF | -157.31±0.28 | 0.8696 | 0.896 | 16.64 | 0 |
| | | AutoNF | -151.73±0.28 | 0.7817 | 0.8758 | 6.41 | +51.36% |
| | Inverse | RQ-C | -157.54±0.28 | 0.8624 | 0.9069 | 13.65 | 0 |
| | | AutoNF | -149.79±0.28 | 0.7848 | 0.8332 | 6.1 | +46.40% |

## 5 DISCUSSION

Normalizing flow is highly parameterized module and designing a flow model and use it for application requires a lot of hands-on experience and domain knowledge. In this paper, we show that the AutoNF framework is very effective in balancing performance-cost trade-offs when building complex flow models. Moreover, although not demonstrated in this paper, the framework could also be used to help decide hyper parameters in complex flow model, e.g. the hidden features and number of bins in the SOTA coupling layer (Durkan et al., 2019). In additional, the proposed optimization method with upper bound can be easily extended to other suitable probabilistic kernels. one example is to identify the best parameterized distribution(s) within a mixture model. We believe our framework will be very useful in many machine learning applications where normalizing flows are needed.

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
