# OpenReview forum: "AutoNF: Automated Architecture Optimization of Normalizing Flows Using a Mixture Distribution Formulation"
_ICLR.cc/2022/Conference — ICLR 2022 Submitted_

### Official Review · Reviewer_bHa6 · 2021-10-31

**Correctness:** 3
**Technical Novelty And Significance:** 3
**Empirical Novelty And Significance:** 3
**Recommendation:** 3
**Confidence:** 4

**Main Review:**

Positive points:
1. In this paper, the authors propose an automated normalizing flow architecture search method which can find the best distribution for each layer from a set of given distribution sequences.
2. When constructing each layer of the model, the authors used a weighted summation of probability density instead of each distribution to ensure the reversibility of the model and the simple calculation.
3. The authors optimized the model via approximate upper bound instead of using KL divergence between the target distribution and the mixed flow, so that the model can go out of the local minimum.
sssssss
Negative points:

1. The novelty of this paper seems limited. The authors directly apply NAS techniques to the field of Normalizing Flow (NF). It would be better to clarify whether there are some technical contributions regarding to the search algorithm.

2. There seem some errors in Eqn. (4). The mixed transformation should be the weighted sum of T_i^j rather than T_i.

3. Eqn. (8) should use the symbol of expectation instead of directly using capital E.

4. As mentioned in Section 3.2, the global minimum may not be the desired architecture. Why can optimizing the upper bound find the desired architecture? It seems that both cases suffer the same issue.

5. Since this paper is essentially a NAS paper, it is necessary to compare the proposed method with existing NAS methods, e.g., DARTS [a], ENAS [b], MNasNet [c].

6. The performance-cost trade-off seems to depend on a parameter lambda that needs to be manually adjusted. Thus, the impact of lambda should be investigated in the experiment section.


Reference:

[a] Darts: Differentiable architecture search. ICLR 2019.

[b] Efficient neural architecture search via parameters sharing. ICML 2018.

[c] Mnasnet: Platform-aware neural architecture search for mobile. CVPR 2019.




**Summary Of The Paper:**

In this paper, the authors present an automated normalizing flow(NF) architecture search method. The method employs a mixture distribution formulation that can construct an optimal flow model with n layers of transformations from the transformation set. Besides, the authors introduce a block-wise optimization method to deal with exponentially growing optimization complexity. In the experiment, the authors proved the effectiveness of the optimization method which via approximate upper bound. And AutoNF has a better performance-cost trade-off than hand-tuned SOTA flow models.

**Summary Of The Review:**

It is unclear the essential technical contributions compared to existing NAS methods.

---

> ### Author Response · Authors · 2021-11-19
> **Thanks and Response to Reviewer 4**
>
> Dear reviewer, thanks a lot for the evaluation and feedbacks of our work. Please see the Global Comments for general questions. The answer to the concerns and confusions you listed can be found below:
>
> (1). As summarized in Global Comment.(1)., differentiable architecture optimization for NF has two unique challenges: the invertibility requirement for transformations and convergence to mixture models. AutoNF is the first work that utilizes a mixed-distribution formulation to construct the valid mixed architecture and enables differentiable architecture optimization.
>
> The proposed Jensen's upper bound has a global optimum as a discrete flow model. Our rigorous proof in the paper shows that under the assumption that global optimum is reachable, optimizing the upper bound will converge to the best discrete flow model and can prevent the potential performance drop when solving the original optimization problem.
>
> (2) & (3). Thanks for pointing out these typos. We will fix them in the revised version.
>
> (4). We need to clarify that our upper bound cannot avoid convergence to local minimum. If the upper bound is non-convex, convergence to global minimum is not guaranteed. For discussion, we assume we can reach global optimum in the end.
>
> As demonstrated in Global Comment.(2)., the global optimum of Jensen's upper bound is the best discrete flow model while the global optimum of the original loss can be a mixture model. For instance, by optimizing original loss, we may result in: $p(x)=0.45\cdot p_{1}(x;\theta_{1}) + 0.55\cdot p_{2}(x;\theta_{2})$. There is no theoretical guarantee that after binarization, $p_{2}(x;\theta_{2})$ will perform better than $p_{1}(x;\theta_{1})$. By optimizing the upper bound, for this example, we will always converge to either $p_{1}(x;\theta_{1})$ or $p_{2}(x;\theta_{2})$, whichever is better.
>
> (5). The formulation of AutoNF is motivated by DARTS which shows better efficiency and results compared with the traditional RL-based or evolutionary-based counterparts. AutoNF shows novel contributions for differentiable architecture search with the mixed-distribution formulation and Jensen's upper bound.
>
> Due to the scope of the work, we leave the comparison with ENAS and MNasNet to the future works. RL-based NAS methods do not face the invertibility constraint and are worth applying to NF models. We are also very interested in exploring one potential limitation of ENAS for NF models: the parameter sharing technique in ENAS restrict the optimization to single cell or fixed number of nodes. The interaction between different flow layers is an open question and the search space to discover such pattern can be large. Whether ENAS can still preserve its efficiency and performance with this large graph remains an interesting research target.
>
> (6). The cost regularization term $\lambda$ is an empirical term and we tune it so that the loss and regularization term can be equally updated. The $\lambda$ for different datasets are listed in the experimental setting in the supplement materials. We have also made some updates to the experimental setup and results, please check the Global Comment.(2). for details.

---

### Official Review · Reviewer_ZCmD · 2021-11-02

**Correctness:** 3
**Technical Novelty And Significance:** 2
**Empirical Novelty And Significance:** 1
**Recommendation:** 5
**Confidence:** 2

**Main Review:**

[Strength]

1. This is the first work using NAS to optimize flow models

2. Although the proposed method is based on DARTs, it requires some efforts to make it work on flow models, such as distribution mixture.

[Weakness]

1. The paper is a bit difficult to follow if the reader is not very familiar with flow model. Especially, the following things need to be further clarified in great detail:

  a) How did you get Eq. (7) from Eq. (6)? In Eq. (7), $W_k=\prod_{i}w_i$, so the right hand side does not contain $k$ at all?

  b) The proof of upper bound optimization should be further clarified. It is hard to follow (at least to me).

2. The upper bound argument seems questionable. While I could understand that we need to binarize the weight $\alpha$ in order to get a simple and valid flow model, I still cannot understand why optimizing the Jensen's upper bound is a good (or better) idea. At least from the experiments, it seems that binarization  is still a necessary step.

3. The proof of Proposition 1 seems questionable. The second $\geq$ is not obvious.

4. The experiments focus on density estimation problems. It is somewhat insufficient. I would expect more real-world applications and comparing to strong baseline methods.

5. Table 1 need improvements. For example, it is better to explicitly align with one cost and then compare the test score. For now it is difficult to compare results since they have different costs.

6. The proposed method almost directly follows DARTS. Although the distribution mixture is novel, it is still more or less an incremental improvement.


**Summary Of The Paper:**

This paper proposes a DARTS-like method for searching automated normalization flow models. Instead of directly using the output ensembles, which leads to infeasible flow models, this work proposed distribution mixture to guarantee that the supernet is always a valid flow model. The upper bound of the loss function is optimized jointly with resource constraints. Experiments on small-to-medium scale datasets valid the effectiveness of the proposed method.

**Summary Of The Review:**

To my best knowledge, this work is the first work aiming to design more efficient flow models via a DARTs-like NAS method. However, there are still many issues to be addressed. This work can be made significantly stronger if the aforementioned issues are well addressed.

---

> ### Author Response · Authors · 2021-11-19
> **Thanks and Response to Reviewer 3**
>
> Dear reviewer, we appreciate your encouraging comments. Please first check the Global Comment for general questions. Here are our replies to your specific questions:
>
> (1). We apologize for the difficulty in reading the paper. Due to the page limit, we removed some of the backgrounds for normalizing flow to supplement to make space for our main contributions.
>
> (1).(a). Recall for $n$ layers and $m$ options. The flow ensemble will consist $m^{n}$ different density functions. The $W_{k}$ corresponds to the weight of a specific density function, which is constructed by selecting one particular transformation in each layer. While the $w_{i}$ is the weight of that particular transformation in the $i_{th}$ layer. Since we don’t know for one density function, which transformation it consists, we simply use $w_{i}$ to indicate the weight of transformation it selects in the $i_{th}$ layer and multiply them together as the final weight for that density function.
>
> (1).(b) Please see answer 3 for clarification.
>
> (2). For discussion purpose, we assume that the global optimum can be reached at the end of optimization.
>
> As we mentioned in the Global Comment.(1).(b)., the global optimum of the original loss can be a mixture model, e.g., $p(x)=0.45\cdot p_{1}(x;\theta_{1}) + 0.55\cdot p_{2}(x;\theta_{2})$. There is no theoretical guarantee that after binarization, $p_{2}(x;\theta_{2})$ will perform better than $p_{1}(x;\theta_{1})$.
>
> The Jensen’s upper bound always has global optimum as the best discrete flow model, i.e., in the previous example, it will always converge to either $p_{1}(x;\theta_{1})$ or $p_{2}(x;\theta_{2})$, whichever is better.
>
> >At least from the experiments, it seems that binarization is still a necessary step.
>
> You are correct that we still apply binarization at the end of optimization, this is only because at that time, updates to weights are slow: the weights are approaching binary values (e.g., 0.9998 vs 0.0002) but still not 1 and 0. However we are confident this is the right approach since this is the desired normal flow model. In contrary, optimizing the original loss might converge to weights as 0.45 vs 0.55. Binarization for these final weights lacks a clear theoretical support.
>
> (3). The second $\ge$ symbol comes from eq.(11), we apologize for the typo and the $\theta_{k}$ in eq.(11) should be $\theta^{*}_{k}$. Combining eq.(10) and eq.(11), we can prove that the global optimum of the upper bound is always a density with optimal parameter which has minimum NLL among all possible density functions.
>
> (4). The key contributions of this work are the mixed-distribution formulation and the proposed upper bound based on Jensen's inequality. Density estimation and performance-cost trade-off are used to demonstrate the validity for one application of our framework. Meanwhile, AutoNF is a very flexible framework and is not limited to density estimation only.  We will explore the application to optimize flow-based generative models or variational inference in the future work.
>
> (5). Based on your suggestions, we have updated the experimental settings and results, as well as adding stronger comparison baseline in the previous literature, please check Global Comment.(2). for details.
>
> (6). As summarized in Global Comments.(1)., our contribution are two folds: first, to adapt to the invertibility of NF, we propose the first work of using a mixed-distribution formulation to construct valid mixed flow architecture. Second, Jesen’s upper bound is proposed to provide a theoretical better global optimum. This convergence guarantee to discrete normal architectures prevents the potential performance drop when applying binarization to the searched mixture model.

---

> > ### Comment · Reviewer_ZCmD · 2021-11-23
> > **Still some questions in clarity and the upper bound proof**
> >
> > Thank your for the reply! After reading the feedback, I still have concerns about the proofs and experiments.
> >
> > 1) I do appreciate the "Additional Background" in the Supplementary. However, Eq. (5) to Eq. (12) still need more detailed explanation. For example, Eq. (7) is still very confusing.  I understand that the subscript $k$ denotes different paths so there is $m^n$ paths in total. But it is confusing that $W_k$ is independent to $k$. If $w_k$ is independent to $k$, so why $K_k$ is inside the $\sum_k W_k \cdots $ rather than $W_k \sum_k \cdots$.
> >
> > 2) I still have concerns about the correctness and the significance of the proof.
> >
> > a) First, in Eq. (9), after using Jensen's upper bound, the optimization degrades to simple maximum selection if there is no constraint. This simplification might be too strong and over-simplified.
> >
> > b) When there are constraints, the upper bound given by Eq. (9) still cannot guarantee to convergence to {0,1} discrete solution. The authors said that after sufficient training, the path selection weight $w_i$ is very close to {0,1} distribution. However, this is only theoretically guaranteed if the optimization problem is simplex programming. The authors must prove that in the constrained programming, the optimal solution to the Jensen's upper bound is still discrete. If theoretical proof is difficult, please at least prove this numerically.
> >
> > c) Again, even if Jensen's upper bound did converge to {0,1} solution, there is no guarantee that this solution is better than naive binarization method. This is because the solution of Jensen's upper bound can be very far from the original optimal solution. So I am not sure choosing Jensen's upper bound as objective function is indeed a *better* idea. Sparse solutions can be obtained in many other ways, for example using {0,1} penalty terms.
> >
> > 3) The authors add more experiments. But what I really expect are comparisons in real-world tasks. I do acknowledge that conducting such experiment might be unrealistic during rebuttal window. These experiments are better to be ready in the first submission.
> >
> > 4) I have no question about the novelty of this work. But  I do concern about whether the contribution in novelty is sufficient and significant. It seems that the flow normalization trick is the only key innovation of this work, which is straightforward.

---

> > > ### Author Response · Authors · 2021-11-24
> > > **Thanks and Clarifications to Concerns of Reviewer 3**
> > >
> > > Dear reviewer, thanks for your reply. Please find the answers to your concerns below:
> > >
> > > **(1).** We apologize there’s a typo in eq.(7): $\sum_{k=1}^{m^{n}} W_{k}\cdot p_{i}(x;\theta_{i})$ should be$ \sum_{k=1}^{m^{n}} W_{k}\cdot p_{k}(x;\theta_{k})$.
> > >
> > > The final flow ensemble $p_{Mix}$ is a mixture model containing $m^{n}$ density functions and each $W_{k}$ is the weight of the $k_{th}$ density function. Suppose this $k_{th}$ transformation is constructed by selecting a specific transformation in each layer, then, $W_{k}$ should be the product of the weights of all selected transformations. For simplicity, we use $W_{k} =\prod_{i=1}^{n}w_{i}$ without explicitly showing which transformation is actually chosen in each layer.
> > >
> > > As a simple example: for a two-layer flow model with options $T^{1}, T^{2}$. $w_{i}^{j}$ donates the weight of the $j_{th}$ transformation in the $i_{th}$ layer. The final flow model is a mixture model summed by 4 density functions: $p_{Mix}=W_{1}\cdot p_{1}(x)+ W_{2}\cdot p_{2}(x)+ W_{3}\cdot p_{3}(x)+ W_{4}\cdot p_{4}(x)$.
> > >
> > > Let $p_{1}(x)=p_{T^{1}T^{1}}(x), p_{2}(x)=p_{T^{1}T^{2}}(x), p_{3}(x)=p_{T^{2}T^{1}}(x), p_{4}(x)=p_{T^{2}T^{2}}(x)$, then we have $W_{1}=w_{1}^{1}\cdot w_{2}^{1}, W_{2}=w_{1}^{1}\cdot w_{2}^{2}, W_{3}=w_{1}^{2}\cdot w_{2}^{1}, W_{4}=w_{1}^{2}\cdot w_{2}^{2}$.
> > >
> > > **(2).(a)** The optimization target is to find the best valid flow model. Our proof shows that at these desired target points, the upper bound is the same as the original loss, for our search target, the search space is not reduced and there’s no over-simplification.
> > >
> > > **(2).(b)**. Our conclusions still hold in the constrained setting: the proof can be found in proposition 3:
> > >
> > > [Proof in constrained programming](https://drive.google.com/file/d/1CMSKA6Ppzn9tAiIIefqdOokjQIXJ5QAK/view?usp=sharing)
> > >
> > > **(2).(c)** Our proof is based on the assumption that **the global optimum is always reachable at the end of the optimization**. The global optimum of the Jensen’s upper bound is guaranteed to be the best valid flow model. On the other hand, the global optimum of the original loss can be a mixture model. The nearest binarization, though close to this mixture model, is not the always the best discrete model.  This is supported by our experiment result in Figure.3 in the paper.
> > >
> > > As a simple example, suppose we have a target distribution as $p^{*}(x)=0.3\cdot p_{1}(x) + 0.7\cdot p_{2}(x)$ and we want to find whether $p_{1}(x)$ or $p_{2}(x)$ best approximates the target distribution (has smallest KL divergence).
> > >
> > > Optimizing the original loss will end up in a mixture model as $ p(x)=0.3\cdot p_{1}(x) + 0.7\cdot p_{2}(x)$ and after binarization, $p_{2}(x)$ will be selected. However, there is no guarantee that this $p_{2}(x)$ has a smaller KL divergence. On the other hand, if we optimize the Jensen’s upper bound, it will converge to either $p_{1}(x)$ or $p_{2}(x)$, whichever has smaller KL divergence.
> > >
> > > Sparse regularization such as entropy-based term can indeed help with convergence to discrete models, however, it does not provide a guarantee that the final solution is the **best** discrete model and requires fine parameter tuning.
> > >
> > > **(4).** We believe with the clarifications on the Jensen's upper bound, our contributions are not only the first work to present a new formulation to enable efficient differentiable architecture optimization for NF, but also providing a rigorous proof that optimizing the upper bound guarantees convergence to the best discrete model.  Both are vital to correct and efficient optimization of the architectures for NF models.

---

> > > > ### Comment · Reviewer_ZCmD · 2021-11-24
> > > > **The theoretical part is still in doubt**
> > > >
> > > > Thank you for updating the feedback.
> > > >
> > > > After reading the new feedback, I still feel that there are more problems in the mathematical formulation and the proof. It is OK if this work comes without any theoretical analysis but since it claims its contribution in theory, all proofs and conclusions must be examined carefully.
> > > >
> > > > 1) The notation is still confusing. After the authors corrected the typos and illustrated with two layer examples, I can now finally understand the meaning of Eq. (7). However, there are more similar issues to be fixed, such as $w_i$ must be denoted as $w_i^j$ for each layer $j$.
> > > >
> > > > 2) Proof is difficult to follow. Most steps do not have any explanation. Combining with the notation issue, the readability is too poor.
> > > >
> > > > 3) The constrained proof still seems questionable. Due to the aforementioned poor readability, it is hard to help the authors to fix the proof with minimal revision. In general, the constrained optimization is a non-linear programming problem. The convergence proof is usually built on KKT condition, Hessian matrix eigenvalue, etc., which usually takes a few pages. The authors's proof directly gives the lower bound in a few lines, leaving many unaddressed questions in the proof. For example, In the proof of Prop 3, why the inference cost can be decomposed in this way? Why in Eq. (5~7), the cost term $C_{pk}$ does not depend on $\theta$ or $p^*$ ?  I seriously doubt how many readers can actually follow the current proof.
> > > >
> > > > 4) The `over-simplified` refers to simplify the non-linear programming to a trivial problem, not reducing the search space. The original problem is an NP-hard {0,1}-programming problem, so it should take great caution when using a polynomial method to approximately solve it. For example, it is necessary to prove that the approximated solution is close to the original optimal solution with high probability. Such bound is not given in this work. So the authors slightly over-claim their results. It is more appropriate to only claim the empirical improvements and acknowledge that there is no  theoretical guarantee of the approximation quality.

---

> > > > > ### Author Response · Authors · 2021-11-25
> > > > > **Thanks to Reviewer 3 and Revised Proof**
> > > > >
> > > > > Dear Reviewer, we appreciate for your time and comments on the previous reply. We must apologize that the previous version is not clear and difficult to follow. Based on your suggestions, we've made a revision on the proof, with a more clear definition of the optimization problems. We've attached a revised version as follow:
> > > > >
> > > > > [Revised Proof](https://drive.google.com/file/d/1qW2C79WdX5v12l7D1uz8M7dPV23IWO5J/view?usp=sharing)
> > > > >
> > > > > **The key idea can be summarized as:**
> > > > >
> > > > > The original optimization problem ($OP_{1}$) is a difficult {0,1} programming problem with the constraint that optimal solution must be a discrete flow model. Binarization or entropy based regularization can help with convergence to a discrete model but does not guarantee its optimality.
> > > > >
> > > > > We propose to solve an equivalent optimization problem ($OP_{2}$), which does not have the previous constraint. We prove that if $OP_{1}$ has a unique optimal solution, solving $OP_{2}$ can lead to exactly the same optimal solution in $OP_{1}$.
> > > > >
> > > > > For pure performance optimization without cost regularization, the proof still holds by simply letting $\lambda$ to 0.
> > > > >
> > > > > **Answers to specific questions:**
> > > > >
> > > > > (1). We've fixed this issue in the revised proof to make it more clear.
> > > > >
> > > > > (3). The cost of each transformation is pre-extracted and depends only on the transformation type, but not on its model parameter.

---

### Official Review · Reviewer_ucQj · 2021-11-02

**Correctness:** 2
**Technical Novelty And Significance:** 3
**Empirical Novelty And Significance:** 3
**Recommendation:** 5
**Confidence:** 4

**Main Review:**

Pros:

Overall the paper is well written and easy to follow. The main idea is to adapt the differentiable formulation for the NAS towards normalizing flow architecture optimization, and the authors made interesting theoretical contributions for reformulating the mixture-of-experts setup towards normalizing flow models, as well as for proposing novel optimization strategies. Overall the proposed method is sound and coherent.

Cons:

- The experimental evaluation seems relatively weak. Though the optimized architectures seem to consistently have lower train, forward and inverse costs, the test performance is mixed, and in many scenarios worse-performing than the manually design architecture.
- The search space that the authors experimented with is extremely limited, limited to only planar flows and radial flow, with the weights between the two transformations being the only architectural hyperparameter learned, not accounting for other hyperparameters such as number of stacked flows, the network complexity (number of feature layers, etc) for each network, which has little generalizability to more modern and useful architectures (e.g., RealNVP, GLOW, FFJORD).
- On the novelty side, though the authors made problem-specific adaptations for the differentiable architecture search algorithm for normalizing flows, the main idea is very much adapted from Liu et al. 2019 and somewhat marginally novel.


**Summary Of The Paper:**

In this paper, the authors proposed to adapt a differentiable architecture search formulation (Liu et al., 2019) based on learned weighting of an ensemble of modules to automated search for Normalizing Flow architectures. The authors made several adaptations to the original approach for the normalizing flow problem, due to the invertibility constraints that prevent direct linear summation of different transform operations. Furthermore, the authors proposed to optimize the full network using an approximated upper bound of the KL divergence, instead of directly optimization. The authors proposed two methods to decompose the optimization problem: grow method, which is more straightforward and greedy, and block method, that alternatively adjusts each block. The authors experimentally compared their proposed method with manually specified architectures across various datasets, including POWER, GAS, HERMASS, MINIBOONE and BSDS300. The results seem mixed, as the searched model outperforms the manual model in some contexts but not others.

**Summary Of The Review:**

As a summary, the overall novelty of the proposed approach is rather incremental, the search space is too restrictive and limited to be practically useful for designing practical NF architectures (only searching for a binary choice between two architectures, not accounting for other more complex bijective transformations), and the experimental results are insufficient to illustrate the usefulness of the proposed approach for improving NF architectures.

---

> ### Author Response · Authors · 2021-11-19
> **Thanks and Response to Reviewer 2**
>
> Dear reviewer, thank you for your time spending on the review and the feedbacks, please check Global Comment for general questions and our answers to your specific questions are as follows:
>
> (1). AutoNF is a highly flexible framework based on the users’ need. In our paper, we define the performance and cost trade-off as the optimization target and evaluate the searched models. However, if the user focuses only on the performance, we are also able to optimize the flow architecture which have better NLL. Please refer to answer (2) for an example.
>
> (2). AutoNF is not restricted to only optimizing within different flow types. The hyper-parameters for a specific flow can also be optimized by this framework. The search space of the framework can be very flexible. As an example, for rational quadratic autoregressive flow (RQ-AF), we evaluated four sets of hyper-parameters with POWER dataset with four layers:
>
> Flow setting 0 (literature setup): # bins: 8 # hidden features: 256
>
> Flow setting 1: # bins: 8 # hidden features: 128
>
> Flow setting 2: # bins: 8 # hidden features: 512
>
> Flow setting 3: # bins: 4 # hidden features: 256
>
> Flow setting 4: # bins: 16 # hidden features: 256
>
> The test NLL results are shown as below:
>
> | Architecture | Test NLL |
> | ----------- | ----------- |
> | [0,0,0,0] (literature baseline) | -0.445±0.013 |
> | [1,1,1,1]	| -0.428±0.013 |
> | [2,2,2,2]	| -0.438±0.013 |
> | [3,3,3,3]	| -0.342±0.013 |
> | [4,4,4,4] (Best setting) | -0.482±0.013 |
> | [1,4,4,4] (searched setting) | -0.479±0.013 |
>
> The result shows that our AutoNF framework has a very flexible search space and can get even further performance improvement than manual settings by searching the hyper-parameters.
>
> (3). The novelty of the paper have been summarized in the Global Comment.1. Differentiable architecture optimization for NF has two unique challenges: the invertibility requirement of transformations and convergence to mixture models. AutoNF is the first work that enables the construction of valid differentiable search space for NF with the mixed-distribution formulation. The Jensen's upper bound provides a rigorous theoretical guarantee of having global optimum as desired discrete flow model.

---

### Official Review · Reviewer_gQWy · 2021-11-03

**Correctness:** 3
**Technical Novelty And Significance:** 4
**Empirical Novelty And Significance:** 4
**Recommendation:** 8
**Confidence:** 3

**Main Review:**

Strengths

The problem considered in this work is novel and important for the community that works with flows. The problem is significantly more challenging than standard architecture search. The proposed solution is well-motivated. The theoretical claims seems to be correct. The flow of the paper is easy to follow and each step of the proposed approach is justified. The novelty and contribution of the paper is high in my opinion. The idea of using mixing the probabilities and the application of upper bound instead of direct optimisation is non-trivial.


Weaknesses

The empirical evaluation of the proposed method should be more extensive. The selection manual flow baseline is a bit tricky to be. How far is the manual approach from the optimal combination of transformations on validation set? It would be also beneficial to create the baseline where the transformation are selected randomly. Such a baseline would deliver the information what is the gain on NLL with respect to the random approach. I would suggest also to take under consideration evolution approach that optimize the binary vectors that stays behind the selection process as reference method.

My second concern is about the generalisation of the proposed approach to other types of flows. The approach seems to be generic and scale to any possible transformations, but the experimental evaluation is mainly focused on autoregressive flows. Is it possible to adopt that approach to various types of layers that represent the dynamics in CNF? It would be also beneficial to see what is the quality of that approach for various CNF layers in experimental part.

The third concern is about the limitation of the approach due the fact that complexity grows exponentially and some decomposition methods are essential to apply architecture search effectively. For this part it would be interesting how much we loose during the decomposition process.

**Summary Of The Paper:**

This work provides the novel approach for searching flow architectures. Compared to the standard approaches used to find the best deep architecture in standard network networks this problem is more challenging due to the need of invertible transform and requirement for determinant of the Jacobian to be easy to calculate. To solve the problem the authors propose to apply the weighting for candidate transformations. In order to enforce invertible properties of the model the authors suggest to use the mixing distribution approach instead of mixing the base transformations. They formulate the problem of learning best weights and show that optimal solution for soft weights is not optimal for binarised versions. Therefore they propose to optimize the upper bound of the proposed loss function instead. Further they show how to deal with the problem with larger number of layers. Some experiments are also performed to show the quality of the approach with respect to the baseline that is expert-based selection.

**Summary Of The Review:**

Concluding, the problem consider by the authors is novel and worth investigating, the approach seems to be adequate, the evaluation is limited but I think the paper is worth accepting.

---

> ### Author Response · Authors · 2021-11-19
> **Thanks and Response to Reviewer 1**
>
> Dear reviewer, thanks for your reviews and acknowledgement to our work. Please see our responses below:
>
> (1). Based on your suggestion, we add additional flow setup and comparison metric as in Global Comment.2.
>
> (2). AutoNF is motivated by the invertibility requirement when constructing valid mixed flow models in discrete transformations. For continuous normalizing flow (CNF), there’s no invertibility constraint when we try to mix different infinitesimal dynamics together, e.g., for $z_{t1}=z_{t0} + \int_{t0}^{t1}w_{1}\cdot g_{\theta_{1}}(z_{t},t) + w_{2}\cdot g_{\theta_{2}}(z_{t}, t)$, the inverse is $z_{t0}=z_{t1} + \int_{t1}^{t0}w_{1}\cdot g_{\theta_{1}}(z_{t},t) + w_{2}\cdot g_{\theta_{2}}(z_{t}, t)$. The Jacobian determinant of the mixed dynamics can also be estimated with [1].
>
> While we can still apply the formulation of AutoNF without any issue to CNF, it is not necessary since the invertibility constraint or complex Jacobian determinant calculation does not exists. The mixed-transformation formulation might be more preferable for CNF models.
>
> (3). To our knowledge, the interaction between different flow layers is still an open problem. Although currently, there are some manual discovered patterns, e.g., alternating linear permutation and autoregressive layers, the optimal sequence of flow layers is still an open problem. The decomposed method may not be able to find global optimum in this greedy manner. For deep flow models, we believe the block method is still needed to allow the previous optimized architectures to adjust themselves.
> As a supplement, at least for simple search space, decomposition method does not have a significant impact. For instance, when searching a 4-layer architecture between planar and radial flow for POWER dataset, optimizing the architecture one layer by one layer and optimizing the whole architecture at once will lead to same results (four planar layers). The greedy method can even show some speed up in search time.
>
> Reference:
>
> [1] FFJORD: Free-form continuous dynamics for scalable reversible generative models, ICLR, 2019.

---

### Author Response · Authors · 2021-11-19
**Global Comment**

Dear reviewers, thanks for the reviews and feedbacks.  We’ve summarized the common comments and concerns in two aspects

**(1) Novelty of the Work (R2.3, R3.6, R4.4).**

The contributions of the work are two folds:

**a.** Novel formulation for differentiable architecture search in normalizing flow (NF).

This is the first work targeting architecture optimization in NF with a rigorous mathematical formulation. Invertibility requirement of transformation in NF prevents direct application of DARTS-like algorithms. The proposed formulation, which applies weighted combination of distributions instead of weighted combination of transformations makes it possible to construct a valid mixed flow model without violating invertibility constraint.

**b.** Theoretical guarantee of convergence to best discrete architecture with proposed Jensen's upper-bound.

Optimizing the mixed flow model directly can lead to undesired mixture models and finding the nearest binarization may cause performance drop. Our proposal ensures that under the assumption global optimum is reachable, and optimizing the Jensen’s upper bound will guarantee the best discrete flow model with a rigorous proof: at discrete architecture points, the upper bound and the original loss are equal to each other, and the global optimum of the upper-bound is always the best discrete flow model.

**(2) Improved Evaluation Metric (R1.1, R2.1, R3.4,5).**

**a.** Reference Designs and New Search Space:

Instead of using manually designed flows following structures of previous SOTA literatures as in the original manuscript, we have revised the manuscript to compare with the reported SOTA designs in published references to provide a stronger baseline. We use the rational quadratic flows family, including rational quadratic autoregressive flow (RQ-AF) and Rational quadratic coupling layer (RQ-C) in [1] which have top 2 performance as the baseline. For fair comparison, we use RQ-AF as the baseline when emphasizing forward cost since it has better performance and use RQ-C as the baseline when emphasizing inverse cost since RQ-C has significantly lower inverse cost.

The search space of AutoNF is also no longer constrained to architectures with fixed number of layers, which allows the discovery of the flow models with fewer layers and better performance-cost trade-off.

**b.** New Evaluation Metric:

Evaluating the performance-cost trade-off is an open question in NF, we propose to use a new metric to address the difficulty of negative log-likelyhood (NLL). NLL is a common measurement for density estimation (lower, the better), however, the order of magnitude of NLL is different across different datasets and it is not suitable to use percentage difference to measure how a model is exactly better than another.

We proposed to apply density and coverage [2] to evaluate the performance of NF models. Density and coverage are recently proposed method to evaluate the sample quality of generative models. The density metric reflects the fidelity of the model and is consistent with NLL metric. Across different datasets, density and coverage are at the same order of magnitude and allows evaluation of architecture across datasets.

To evaluate the performance-cost trade-off, we define a figure of merit (FOM) as FOM = cost reduction% + density drop% compared to reference SOTA designs. In principle, the weight of the two terms can be manually adjusted to reflect the importance. For demonstration purpose, we use the equally weighted summation.

**(3) Additional Experiments and Results:**

**a.** Five different datasets are used to evaluate the performance-cost trade-off.

[Result Table of Additional Experiment a.](https://i.imgur.com/hir4f0z.png)

**b.** To further demonstrate the generality of AutoNF, we present its ability to tune hyper-parameter of a specific flow model.

Flow 0 (literature setup): # bins: 8 # hidden features: 256

Flow 1: # bins: 8 # hidden features: 128

Flow 2: # bins: 8 # hidden features: 512

Flow 3: # bins: 4 # hidden features: 256

Flow 4: # bins: 16 # hidden features: 256

The test NLL results are shown as below:

| Architecture | Test NLL |
| ----------- | ----------- |
| [0,0,0,0] (literature baseline) | -0.445±0.013 |
| [1,1,1,1]	| -0.428±0.013 |
| [2,2,2,2]	| -0.438±0.013 |
| [3,3,3,3]	| -0.342±0.013 |
| [4,4,4,4] (Best setting) | -0.482±0.013 |
| [1,4,4,4] (searched setting) | -0.479±0.013 |

**c.** Analysis:

We show that across 5 different datasets, out AutoNF can reach to up to 3.66 X cost reduction and up to 75.2% improvement in FOM compared with SOTA literature results.

Except for improvement in performance-cost trade-off, AutoNF can also be applied to find better hyper-parameter settings in flow models and provide further performance boost compared with manually selected setting in SOTA literature.

Reference:

[1] Reliable Fidelity and Diversity Metrics for Generative Models, ICML, 2020

[2] Neural Spline Flow, NeurIPS, 2019

---

### Decision · Program_Chairs · 2022-01-20

**Decision:**

Reject

**Comment:**

This paper proposes two methods to learn the architecture of normalizing flows models; Their framework is inspired by (Liu et al., 2019) which uses ensembles/mixtures with learnable weights for architecture search. The application of these ideas to NFs requires a trivial modification to respect the invertibility constraint; which consists in building a mixture model over all possible sequences of compositions of transformations from a fixed set.

The paper proposes to use an upper-bound to the forward KL instead of the fKL directly. The reasoning is that this will lead to a "pure" model after optimization, that is, the mixture weights will be in {0, 1}. Mathematically, this simply corresponds to treating the mixture as a latent-variable model and performing MAP-inference over discrete latent variables, assuming that all mixture components have the same prior weights in the mixture.

The experimental results across various datasets are very mixed, and the family of transformations considered in the experiments is quite restricted.